# "On-Water" accelerated dearomative cycloaddition via aquaphotocatalysis

Soo Bok Kim[1], Dong Hyeon Kim[1] & Han Yong Bae ⬡ [1] ✉

Sulfur(VI) fluoride exchange (SuFEx) has emerged as an innovative click chemistry to harness the pivotal connectivity of sulfonyl fluorides. Synthesizing such alkylated S(VI) molecules through a straightforward process is of paramount importance, and their water-compatibility opens the door to a plethora of applications in biorelevant and materials chemistry. Prior aquatic endeavors have primarily focused on delivering catalysts involving ionic mechanisms, studies regarding visible-light photocatalytic transformation are unprecedented. Herein we report an on-water accelerated dearomative aquaphotocatalysis for heterocyclic alkyl SuFEx hubs. Notably, water exerts a pronounced accelerating effect on the [2 + 2] cycloaddition between (hetero) arylated ethenesulfonyl fluorides and inert heteroaromatics. This phenomenon is likely due to the high-pressure-like reactivity amplification at the water-oil interface. Conventional solvents proved totally ineffective, leading to the isomerization of the starting material.

The catalytic compatibility of water within aqueous environments has garnered increasing interest as it has not only ecological benefits but also influences vital biological processes such as protein folding, substrate-enzyme interactions, and lipid bilayer formation[1–3]. Despite scientific progress, the complexities behind the profound understating of aquatic catalysis remain elusive. A pivotal breakthrough in synthetic chemistry was the revelation of the water-oil interface[4] as the surface of heightened reactivity, which was termed as the on-water effect[5]. Initially studied by Breslow et al. earlier[6] and more refined by Sharpless et al. later[5], this effect involves immiscible reactants agitated in aqueous suspension. This discovery sparked contemplation owing to the analogy between complex biochemical transformations in water and the transferability of these principles to organic synthesis. Through the years, various aquacatalytic systems have been studied[1–3], including Lewis acids[7], organo-transition metals[8] as well as emerging systems such as Brønsted acid[9] and Brønsted base[10] organocatalysts. These systems remarkably improved reaction kinetics and selectivity via their synergistic interactions with water. Intriguingly, this improvement extends to type III reactions[11], in which the insolubility of both the catalyst and the reactant under aqueous media leads to substantial rate enhancement[12]. This is mainly relevant to addition or rearrangement reactions with negative transition state volume changes ($\Delta V^{\ddagger} < 0$)[13], resembling reactivity acceleration akin to high-pressure conditions[14]. This enhancement frequently results in exceptionally favorable effects, surpassing the outcomes achieved using traditional solvents or neat conditions. This phenomenon is prominent in ionic (involving electron pair) catalysis, particularly in Michael-type reactions with electron-deficient olefinic substrates[10,15]. However, remarkable acceleration of cycloadditions driven by bulk aqueous media employing a combination of water-insoluble substrates and hydrophobic photocatalysts has been absent in visible-light photocatalysis (Fig. 1A)[16].

Sulfur(VI) fluoride exchange (SuFEx)[17] is an emerging class of click chemistry[18] that enables the coupling between electrophilic sulfur(VI) molecules and amines/alcohols. The applicability of SuFEx extends beyond connections involving small molecules and includes biomacromolecules of significance, such as proteins and deoxyribonucleic acid (DNA)[19]. Numerous contemporary catalytic studies have shown the advantages associated with operations conducted in aquatic environments. The most representative functional hub for this purpose is the sulfonyl fluoride group, which has the advantage of being stable in water[20] and tolerant to certain catalytic conditions, thereby enabling the introduction of different catalytic reactions[21]. Ethenesulfonyl fluoride (ESF), a primary modular unit for this purpose[22], has excellent potential for use in medicinal and materials chemistry because various substituents can be introduced at the

[1]Department of Chemistry, Sungkyunkwan University, Suwon 16419, Republic of Korea. ✉e-mail: hybae@skku.edu

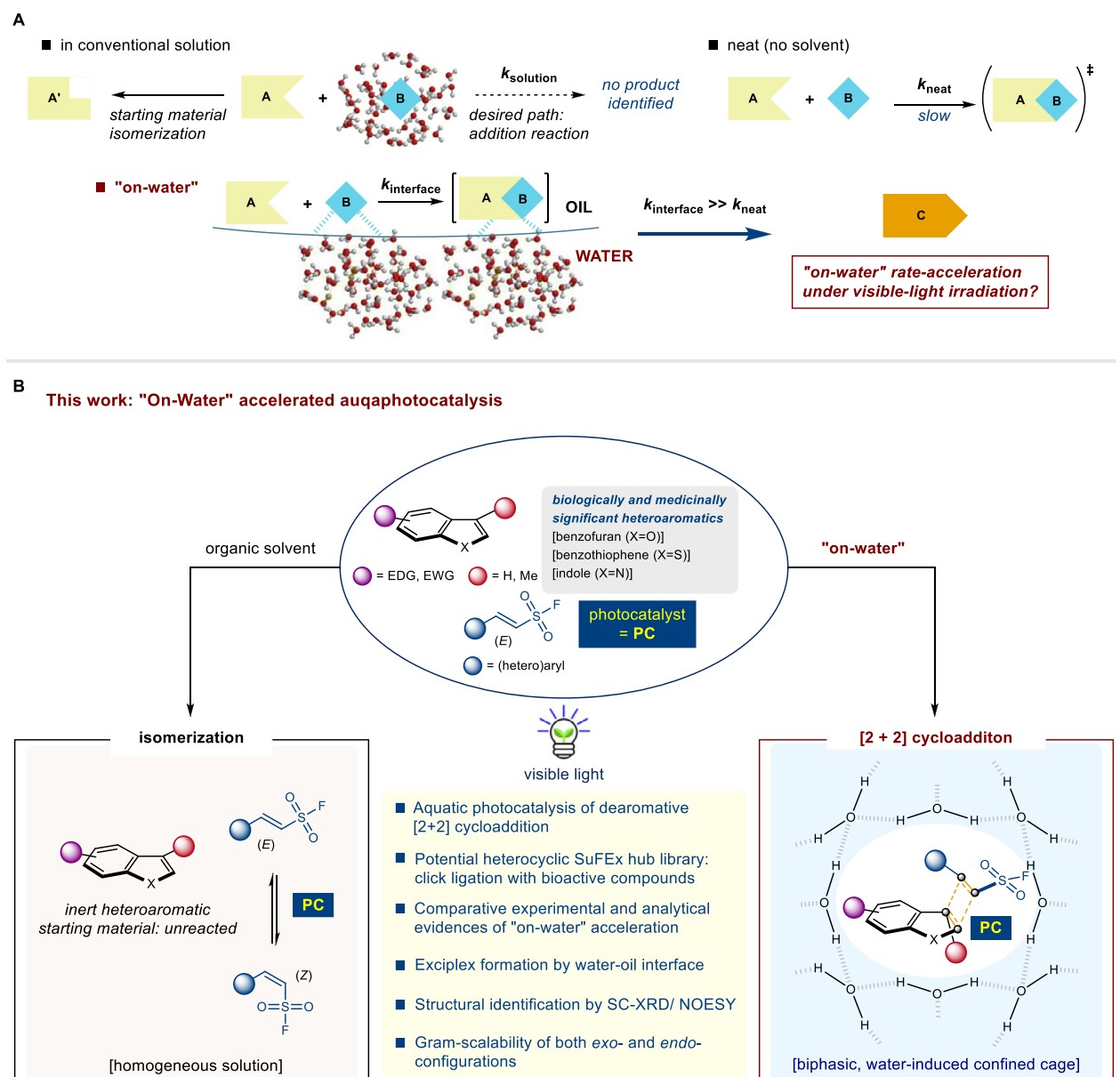

**Fig. 1 | On-water accelerated aquaphotocatalysis of [2 + 2] cycloaddition to access heterocyclic SuFEx hubs. A** Conceptual description of the on-water reaction. **B** Overview of this study. SC-XRD single crystal X-ray diffraction, PL photoluminescence, SuFEx sulfur(VI) fluoride exchange, NOESY nuclear overhauser enhancement spectroscopy.

β-position[23]. Interestingly, β-substituted ESF can tolerate ionic mechanisms involving electron pairs under recently unveiled catalytic conditions, such as a phosphazene superbase[24,25] or *N*-heterocyclic carbene catalysis[26], and can efficiently perform the desired aquatic reactions. However, it is difficult to establish a transformation by suitable photocatalysis because the reaction soon reaches equilibrium via rapid *E/Z* isomerization in an organic solvent[27]. To date, the realization of intermolecular photocycloadditions has been challenging. While investigating the unique acceleration properties of water, we focused on the intriguing prospect of enabling reactions under photocatalytic conditions that are currently unattainable in conventional organic solvents. Notably, there was a lack of documented cases of photocatalysis using aqueous media[16] in the context of on-water acceleration.

Herein, we report an expedited synthesis of heterocyclic SuFEx hubs via aquacatalytic dearomative [2 + 2] photocycloaddition on-

water. We focused on crafting a modular platform of the SuFEx hub library, leveraging the fundamental scaffold as a vital module in biology and medicine. Among biorelevant compounds, benzofuran[28], benzothiophene[29], indole[30], furan, and their derivatives captivated us due to their widespread utilities. Particularly, most benzofuran compounds exhibit potent antitumor, antibacterial, antioxidant, and antiviral activities[31]. The desired cycloaddition products were obtained on a preparative scale from a wide variety of β-(hetero)arylated ESFs and heteroaromatic compounds. This method exhibits dominant head-to-head site selectivity (the carbon-heteroatom linkage and sulfonyl fluoride are aligned in the same direction), producing both *exo*- and *endo*-diastereomers as isolable entities. Notably, the starting materials and the organophotocatalyst were insoluble in water. No detectable byproducts emerged, which could be attributed the high-pressure-like environment created by the water-oil interface within the encapsulating confined cage[32]. Attempts to initiate the reaction under all the

## Table 1 | Optimization of reaction conditions

| Entry | Variation from the standard condition | Recovered S.M. (%) | | Cycloaddition product (%) |
|---|---|---|---|---|
| | | 1a | 1a' | 3aa [exo:endo] |
| 1 | Ru(bpy)₃(PF₆)₂ instead of 4Cz-IPN | >99 | 0 | 0 |
| 2 | Eosin Y instead of instead of 4Cz-IPN | >99 | 0 | 0 |
| 3 | Eosin Y disodium salt instead of instead of 4Cz-IPN | >99 | 0 | 0 |
| 4 | Rose bengal instead of 4Cz-IPN | >99 | 0 | 0 |
| 5 | 9-Mesityl-10-Me-acridinium ClO₄ instead of 4Cz-IPN | >99 | 0 | 0 |
| 6 | thioxanthone instead of 4Cz-IPN | >99 | 0 | 0 |
| 7 | 4DP-IPN instead of 4Cz-IPN | 28 | 7 | 14 [9:5] |
| 8 | fac-Ir(ppy)₃ instead of instead of 4Cz-IPN | 38 | 18 | 44 [31:13] |
| 9 | [Ir{dF(CF₃)ppy}₂(dtbpy)]PF₆ instead of 4Cz-IPN | 0 | 0 | 77 [55:22] |
| 10 | 3.0 equiv. of 2a | 27 | 7 | 63 [47:16] |
| 11 | Without 4Cz-IPN | >99 | 0 | 0 |
| 12 | In dark | >99 | 0 | 0 |
| 13 | 44 W 525 nm green LED instead of blue LED | >99 | 0 | 0 |
| 14 | 52 W 390 nm purple LED instead of blue LED | 0 | 0 | 86 [64:22] |
| 15 | Ambient sun light instead of blue LED | 40 | 40 | 18 [13:5] |
| 16 | Heating at 80 °C instead of light | >99 | 0 | 0 |
| 17 | With TEMPO | 53 | 46 | 0 |
| 18 | None | 0 | 0 | >99 [79:21] |

Reactions were performed with 1a (0.1 mmol, 1.0 equiv.), 2a (0.5 mmol, 5.0 equiv.), and 4Cz-IPN (2.0 mol%) using H₂O (10 L/mol, 1 mL) by irradiated 50 W 456 nm blue LED for 24 h. Product conv. (%) and recovered conv. (%) was determined using ¹H NMR with 1,4-dimethoxybenzene as an external standard. The isolated yields are shown in parentheses.
S.M. starting material, TEMPO (2,2,6,6-tetramethylpiperidin-1-yl)oxyl.

traditional organic solvent conditions proved ineffective, demonstrating the crucial role of on-water reaction conditions in this unique process (Fig. 1B).

## Results

### Optimization of aquaphotocatalytic reaction conditions

In our initial research, the visible light irradiation photocatalytic system (blue, 50 W, 456 nm, Kessil LED lamp) for the reaction between phenyl-ESF (1a) and 2,3-benzofuran (2a) using water as the sole reaction medium were investigated. Table 1 lists the different organic[33] and organometallic photocatalysts[34] that were tested. Notably, well-known

photocatalyst such as Ru(bpy)₃(PF₆)₂ exhibited no reactivity (entry 1). Eosin Y and eosin Y disodium salt, which were excellent catalysts for photocatalytic decarboxylative ESF functionalization gave no product in this reaction (entries 2–3)[35]. Rose bengal, 9-mesityl-10-Me-acridinium ClO₄, and thioxanthone showed no change (entries 4–8). Cyanoarene catalysts such as 4DP-IPN[36] showed very low conversion (14% yield, exo: endo = 9: 5, entry 7). However, improved yields were obtained using fac-Ir(ppy)₃ (44%, exo: endo = 31: 13, entry 8) and [Ir{dF(CF₃)ppy}₂(dtbpy)]PF₆ (77%, exo: endo = 55: 22, entry 9). Surprisingly, organophotocatalyst 4Cz-IPN[37] significantly improved the reactivity (>99% yield, exo: endo = 79: 21, entry 18). Control experiments

revealed that lower amounts of 2,3-benzofuran **2a** (3.0 equiv.) resulted in reduced reactivity (entry 10, 63% yield). No reaction occurred in the absence of a catalyst or visible light irradiation (entries 11 and 12). Different light sources (green, 44 W, 525 nm; purple, 52 W, 390 nm; ambient sunlight) yielded inferior results (entries 13–15). Heating the reaction vessel had no positive effect (entry 16). The addition of the radical scavenger (2,2,6,6-tetramethylpiperidin-1-yl)oxyl (TEMPO) interrupted the reaction and led to interference[33,34], yielding only *Z*-isomer **1a'** (46% yield).

### Effects of reaction media and vinylsulfur(VI) electrophiles

In the optimized model reactions under varying media (Fig. 2A), all conventional organic solvents (acetone, CH$_2$Cl$_2$, DMF, DMSO, Et$_2$O, acetonitrile, MeOH, *n*-hexane, PhMe, and THF) failed to yield [2 + 2] photocycloaddition products, recovering 38–96% yields of unreacted **1a**. Isomerized starting material (*Z*)-phenyl ESF **1a'** was also formed (3–53% yield), without any detectable side products (entries A–J).

A shift to water-rich conditions initiated the desired photocycloaddition reaction. Pure water emerged as the most effective medium, fully converting phenyl-ESF **1a** into the desired products **3aa** [*exo* + *endo*] (entry P, >99% yield). Further control experiments were conducted to provide evidence for on-water acceleration. For example, catalytic amount of amphiphilic surfactant for the reaction was employed to provide a homogeneous micellar environment. The presence of readily available designer surfactants such as TPGS-750-M[38] and SPGS-550-M[39] developed by Lipshutz and co-workers resulted in minimal conversion, which notably decreased the tension at the water-oil interface (entries K and L: 12% and 23%, respectively). In addition, a yield of 84% was achieved in the solvent-free condition (entry M): this observation supports the notion that on-water catalysis inherently exhibits behavior akin to a highly concentrated environment. Following our expectation, heavy water such as D$_2$O gave a relatively lower yield than that of H$_2$O (entry N, 90%). According to the prior deuterium kinetic isotope effect studies conducted by Jung and Marcus, the increased viscosity of D$_2$O can influence shear forces, potentially giving rise to larger droplets[40]. This could result in reduced contact surface areas, and consequently contribute to decreased reaction rates. Sodium chloride, which is a common hydrophobic agent[24], was ineffective in this reaction (entry O).

To establish the indispensable role of the sulfonyl fluoride functional group, we compared and investigated alternative vinyl sulfur(VI) electrophiles. Compounds such as (*E*)-(2-(phenylsulfonyl)vinyl)benzene showed reasonable reactivity (81% yield) but lacked SuFEx click ability. The sulfonyl chloride analogue of the compound **1a** totally decomposed under the established aqueous reaction conditions[17,24]. This observation further emphasized the crucial function of the sulfonyl fluoride moiety in enabling the intended reaction (Fig. 2B).

We compared the reactions under neat and on-water conditions[5,24–26], and reaction profiles were recorded at different time intervals (Fig. 2C). Notably, a significantly boosted reactivity was observed in the initial stages of the reaction under on-water conditions. Within 2 hours, the neat condition yielded 38% of **3aa** [*exo* + *endo*], while the on-water condition exhibited a substantial 80% yield of **3aa** [*exo* + *endo*]. Approaching saturated reactivities were achieved within 12 h, resulting in a 56% yield under neat condition and a 91% yield on-water condition. This stark difference further corroborates the remarkable acceleration effect caused by the presence of the bulk water phase.

### Substrate scope, scale-up synthesis and SuFEx click applications

Under the optimized on-water conditions, various β-substituted ESFs (**1a**–**1q**) and hetero-aromatic compounds (**2a**–**2l**) were subjected in the presence of organophotocatalyst 4Cz-IPN with blue LED irradiation (Fig. 3). *Exo*- and *endo*-structural configurations were determined by 1D NOESY (Nuclear Overhauser Enhancement SpectroscopY) NMR

and single-crystal X-ray diffraction analyses (for detailed data, see Supplementary Information). The reactions proceeded efficiently to afford the desired products in good-to-quantitative yields. Both simple phenyl and β-arylated ESFs of electron-donating/electron-withdrawing substituents coupled with 2,3-benzofuran (**2a**) to achieve successful outcomes (up to 99% yield, products **3aa**–**3oa**). Sulfur and nitrogen incorporated β-heteroaryl ESFs led to the target compounds with yields ranging from 75% to 83% (**3pa**, *exo:endo* = 75:25 and **3qa**, *exo:endo* = 51:49). Different heteroaromatic coupling partners **2**, such as 1-benzothiophene (**2b**), *N*-Boc-protected indole (**2c**), 3-methylbenzofuran (**2d**), 3-methylbenzothiophene (**2e**), furan (**2f**), *N*-Boc-protected-L-tryptophan derivative (**2g**), functionalized benzofuran derivatives (**2h**–**2j**), and functionalized benzothiophene derivatives (**2k** and **2l**) were smoothly converted into their respective products (**3ab**–**3al**) with yields of up to 99%. Interestingly, it is worth to note that in the cases of alkyl substituted heteroaromatics at 3-position such as **2d**, **2e**, and **2g**, *endo* products were obtained in major form (**3ad**, *exo:endo* = 15:85; **3ae**, *exo:endo* = 21: 79; **3ag**, *exo:endo* = 44:56). The reason is unclear at this stage, however, the presence of steric repulsion between the alkyl group at the 3-position of heteroaromatics **2** and the phenyl group of **1a** is believed to favor *endo*-product formation (this is in accordance with a similar situation reported in the Paternò-Büchi reaction, where the intermolecular steric repulsion between the alkyl group on the heteroaromatic and the bulky group on the counterpart was identified as the key factor)[41] In addition, β-unsubstituted ESF were inactive in this catalytic reaction.

Next, preparative-scale synthesis was performed (Fig. 4A). Biphasic reaction mixtures were observed (i) before and (iii) after vigorous stirring, under (ii) blue LED light irradiation setup. Starting material **1a** (1.00 g) on the gram-scale yielded total 1.55 g of the isolable product **3aa** in 95% yield after purification (**3aa**-*exo* (iv): 74%; 1.21 g, **3aa**-*endo* (v): 21%, 0.34 g). The single-crystal X-ray structures of both the *exo*- and *endo*-isomers verified their precise relative chemical structures (CCDC-2279317 and CCDC-2279318, respectively).

Intermolecular SuFEx click reactions[17] were conducted using nitrogen- and oxygen-containing molecules in which the cyclobutane moiety was well tolerated (Fig. 4B). Benzylamine as the donor efficiently provided sulfonamide **4** (step (i), 61% yield). The structurally complex sulfonyl azide **5** was obtained as a clickable hub using TMSN$_3$ and 4-dimethylaminopyridine (DMAP), followed by a traditional Cu-catalyzed azide-alkyne cycloaddition (CuAAC) click reaction[18], affording sulfonyl triazole **6** in 90% yield (steps (ii) and (iii): double-click product). Simple phenolic molecules were converted into sulfonic ester **7** in 86% yield (step (iv)). More importantly, the bioactive compounds were smoothly coupled with phenolic (*sp$^2$*C−OH) and alcoholic (*sp$^3$*C−OH) compounds. Estrone and stanolone transformed to the desired SuFEx click products **8** (64% yield) and **9** (31% yield), respectively (steps (v) and (vi), respectively).

### Mechanistic investigation of on-water accelerated aquaphotocatalysis

We conducted preliminary mechanistic investigations into the on-water acceleration, with a primary focus on delineating the role of water in the reaction process (for detailed analytical data, cyclic voltammetry measurements, UV−vis spectra, and further spectrochemical control experiments, see Supplementary Information). Fluorescence quenching experiments were conducted, and the stacked emission spectra of 4Cz-IPN and phenyl ESF (**1a**) were recorded for various concentrations of 2,3-benzofuran **2a** (0 mM, 50 mM, 100 mM, 150 mM, and 200 mM) in a mixed THF-water medium as depicted in Fig. 5A. Due to the limited solubility of the organic analytes in pure water, co-solutions of THF/water were employed for measurements. The spectra were compared for different THF: water (v/v) medium ratio ((i) 100:0, (ii) 80:20, (iii) 65:35, and (iv) 50:50). Notably,

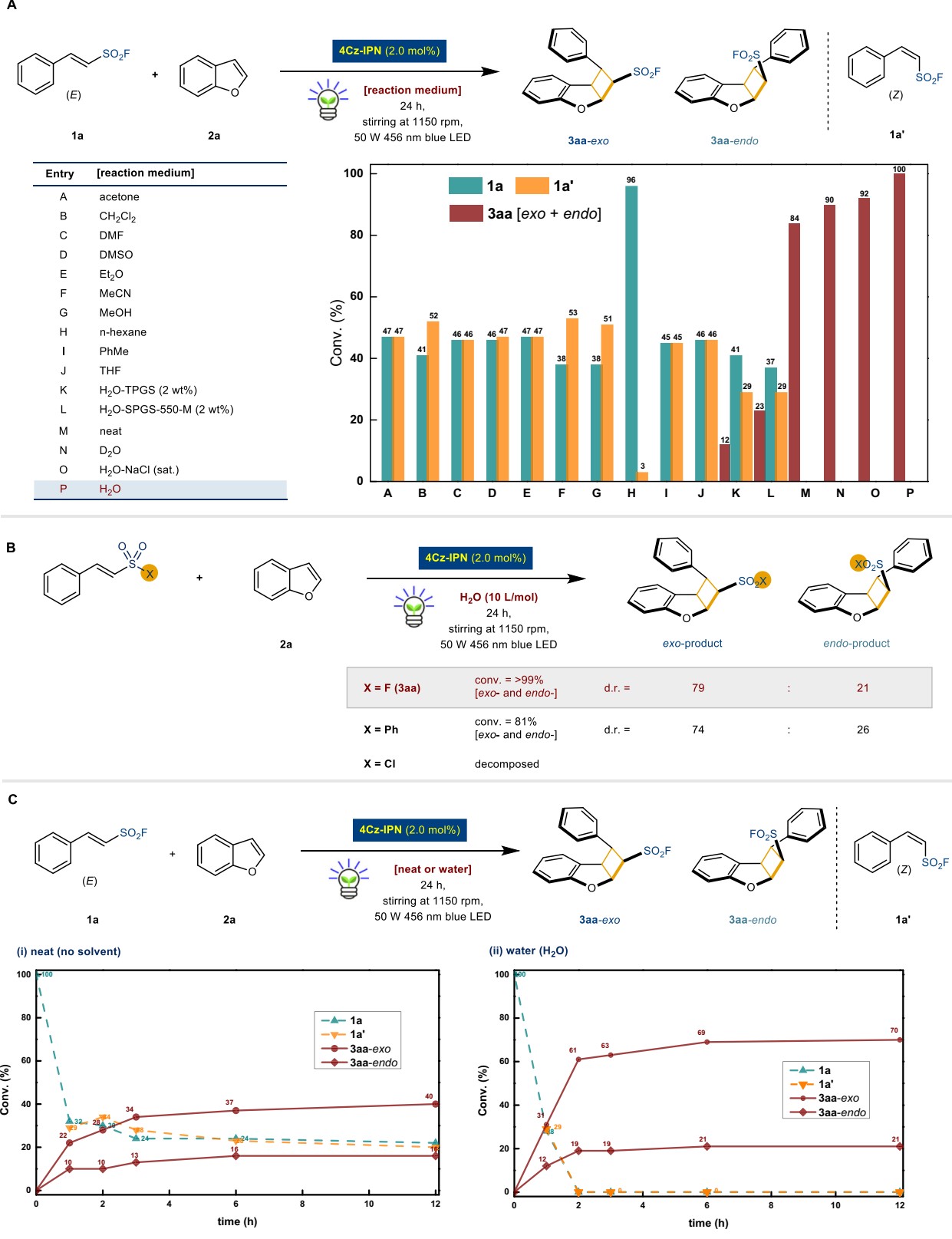

**Fig. 2 | Effects of reaction media and vinylsulfur(VI) electrophiles. A** Detailed reaction medium screening.[a] **B** Reaction stirred with different S(VI) starting materials.[a,b] **C** Reaction profiles under (i) neat condition[a,c] and (ii) on-water condition. [a]Reactions were performed using **1a** (0.1 mmol, 1.0 equiv.), **2a** (0.5 mmol, 5.0 equiv.), and 4Cz-IPN (2.0 mol%) with $H_2O$ (10 L/mol, 1 mL) by irradiation of blue LED (50 W 456 nm) for 24 h. Conversion (conv.) and the diastereomeric ratios (d.r.) were determined by [1]H NMR spectroscopy using 1,4-dimethoxybenzene as an external standard. [b]Different starting materials were used instead of **1a**. [c]In the absence of the reaction medium.

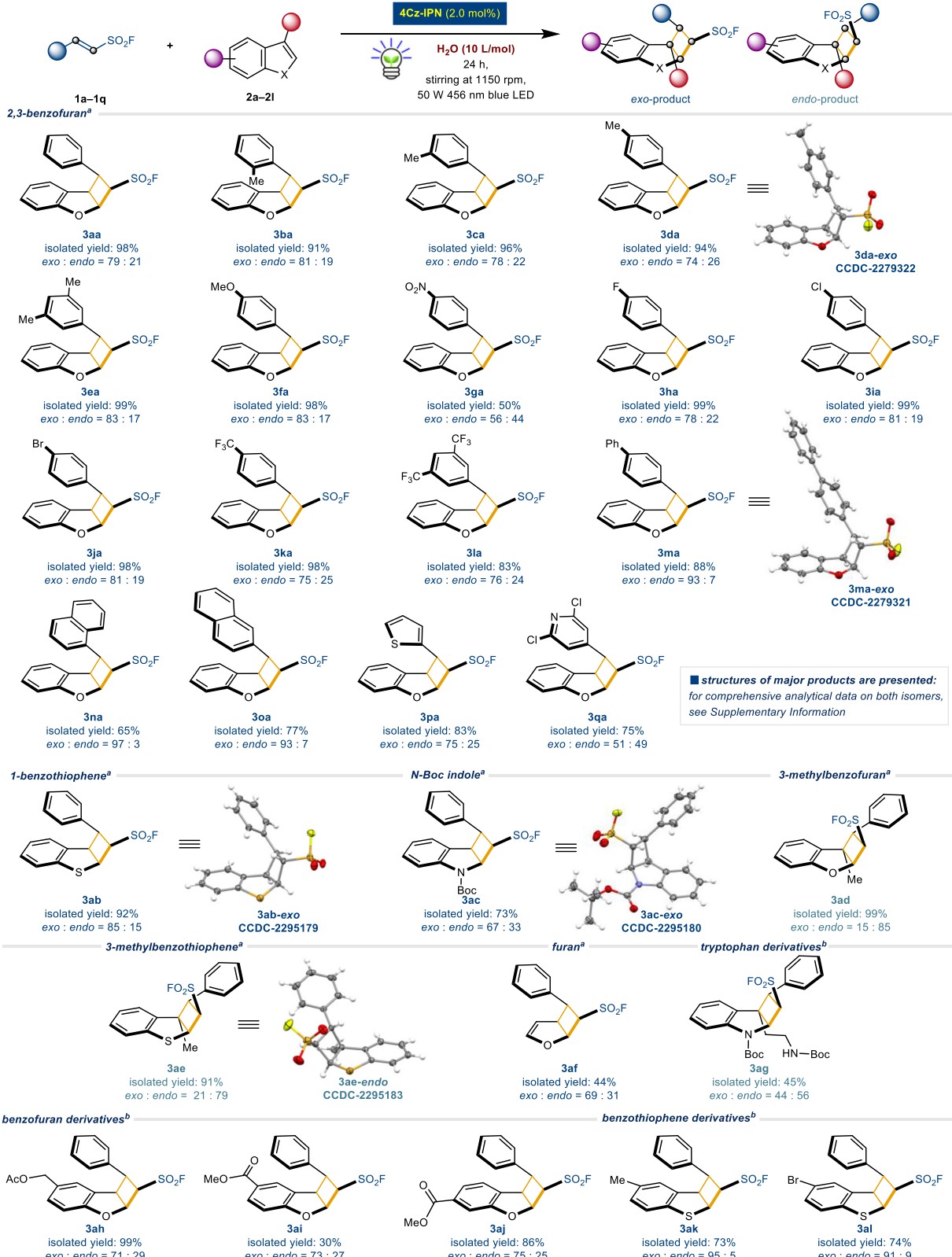

**Fig. 3 | Substrate scope.** [a]Standard condition: Reactions were performed using **1** (0.4 mmol, 1.0 equiv.), **2** (2.0 mmol, 5.0 equiv.), and 4Cz-IPN (2.0 mol%) with $H_2O$ (10 L/mol, 4 mL) by irradiation of blue LED (50 W 456 nm) for 24 h. Diastereomeric ratio (d.r.) was determined by [1]H NMR. [b]Standard condition with $H_2O$ (20 L/mol, 8 mL).

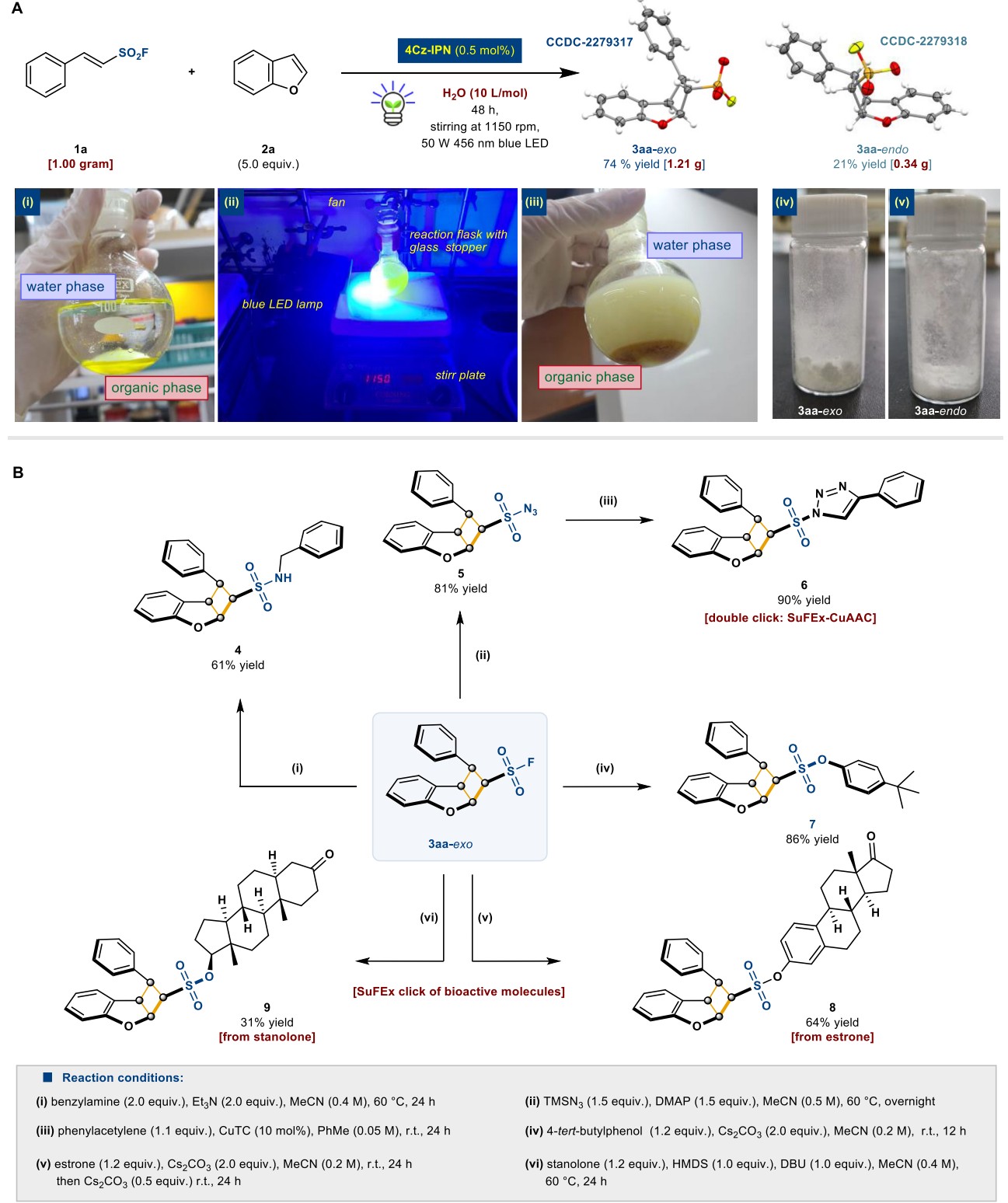

**Fig. 4 | Scale-up synthesis and SuFEx click applications. A** Gram-scale reaction.[a]
**B** Synthetic utilities on SuFEx click reactions. DMAP = 4-dimethylaminopyridine;
HMDS = hexamethyldisilazane; DBU = 1,8-diazabicyclo(5.4.0)undec-7-ene.

[a]Reaction was performed with **1a** and **2a** (5.0 equiv.), and 4Cz-IPN (0.5 mol%)
with H₂O (10 L/mol, 1 mL) by irradiation of blue LED (50 W 456 nm) for 24 h.

pronounced fluorescence quenching was observed as the water frac-
tion increased. This fact underscores the impact of water on fluores-
cence quenching dynamics (Fig. 5B). Stern–Volmer luminescence
quenching[42] analysis demonstrated negligible changes under (i) 100:0,
(ii) 80:20, and (iii) 65:35 THF/water (v/v) ratios, suggesting a nearly

homogeneous solution. Substantial alterations appeared in the pre-
sence of (iv) 50:50 = THF:water, indicating a partly homogeneous
medium. Specifically, under the condition (iv), meaningful yield of
cycloaddition product **3aa** [10%, *exo:endo* = 8:2] was identified. In
addition, when 20 eq. of **2a** was used under identical conditions, **3aa**

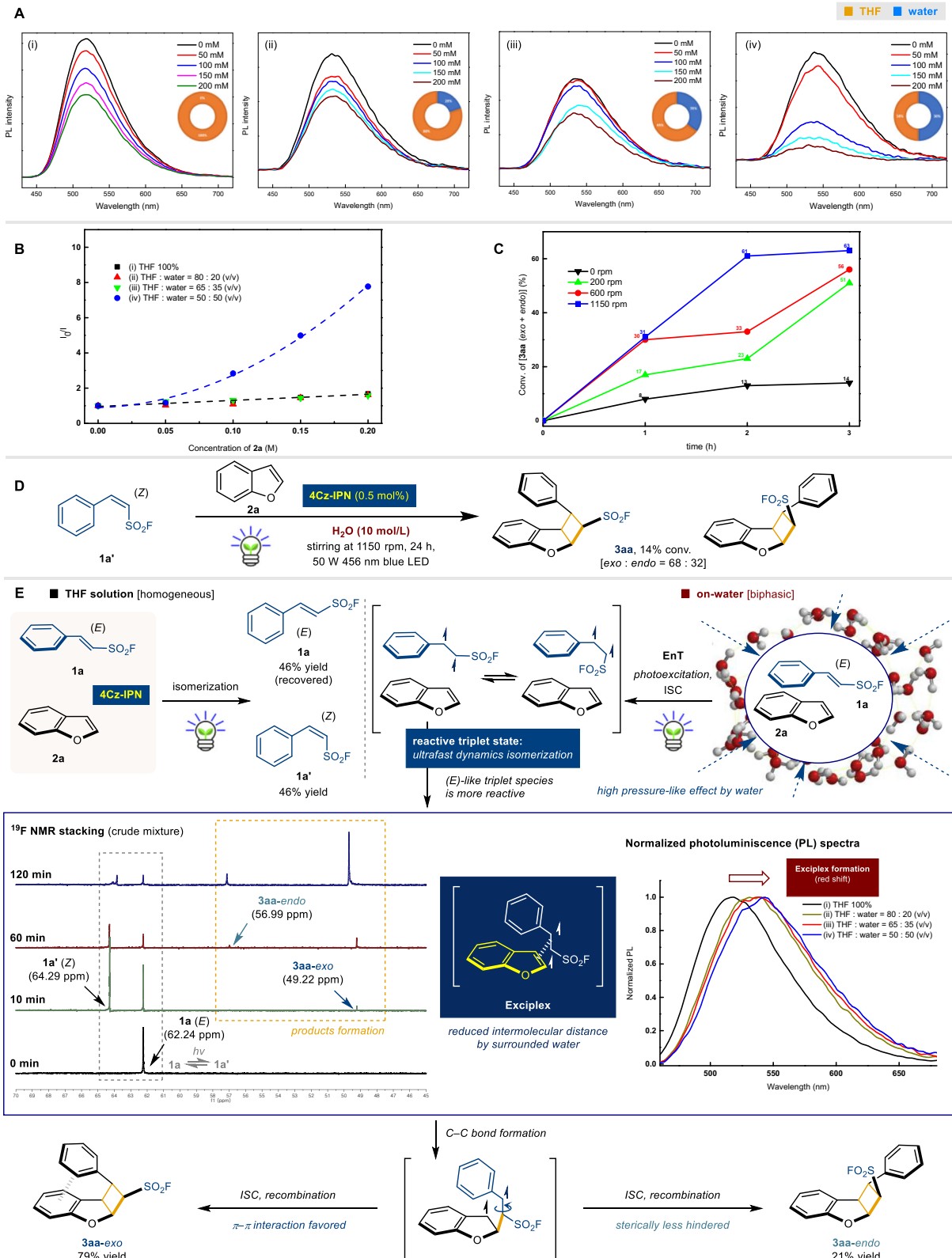

**Fig. 5 | Mechanistic investigation. A** Fluorescence quenching of excited photocatalyst 4Cz-IPN (40 μM) using **1a** (10 μM), with different concentrations of **2a** (0 mM, 50 mM, 100 mM, 150 mM, and 200 mM) under different media (THF/water ratio = (i) 100:0, (ii) 75:25, (iii) 65:35, and (vi) 50:50). **B** Stern–Volmer plot representing the quenching of photocatalyst 4Cz-IPN with **2a** under different reaction media (THF/water ratio = (i) 100:0, (ii) 75:25, (iii) 65:35, and (vi) 50:50). (**C**) Stirring speed-dependent reaction progress of the model reaction at early stage. (**D**) Control experiment using (Z)-phenyl ESF (**1a'**).[a] (**E**) Plausible on-water acceleration mechanism based on the time-dependent [19]F NMR analyses and normalized PL analysis. ISC = intersystem crossing. [a]Reaction was performed under standard conditions. Conversion was determined by [1]H NMR analysis using 1,4-dimethoxybenzene (0.1 mmol, 1.0 equiv.) as an external standard.

[*exo*:*endo* = 25:10] was produced in 35% yield. These facts support that water as a reaction medium is crucial to enhancing this [2 + 2] photocatalytic transformation (a larger proportion of water in the mixed solvent conditions was unsuitable for the measurement; see Supplementary Information for details).

Further insights into the on-water acceleration phenomenon were obtained by studying the impact of mixing speed on the reaction mixture, inspired by prior works[43–45]. The lack of stirring led to sluggish progression due to poor mass transfer, yielding only 13% conversion of **3aa** [*exo* + *endo*] in 2 h. With very gentle stirring at 200 rpm, a slight increase in the product conversion was observed (23% of **3aa** [*exo* + *endo*], 2 h). A gradual increase in the stirring speed to 600 rpm resulted in improved conversion rate (33% conv.). Finally, vigorously stirring of the reaction vessel afforded significantly higher reaction outcome (61% conv.). These observations collectively suggest that the reaction enhancement predominantly occurs at the water-oil interface[4], implying that exposure to a larger interfacial surface area might be a pivotal factor driving efficient reaction outcomes (Fig. 5C).

The reactivity significantly deteriorated when the *Z*-isomer of phenyl-ESF **1a'** was used as a starting material under the standard conditions, providing only 14% conversion of the product with lower *exo*-: *endo*- selectivity (68: 32). This fact supports the constructive [2 + 2] cycloaddition is mainly originated from the *E*-isomer of phenyl-ESF **1a** (Fig. 5D).

The photocatalytic reaction likely follows a Dexter-type energy transfer (EnT) mechanism based on prior [2 + 2] cycloaddition approaches[46,47]. Following photoexcitation to an excited singlet state, the donor 4Cz-IPN undergoes intersystem crossing (ISC), transitioning from a singlet to a triplet state. This triplet state is expected to exhibit significant lifetime, allowing ample time for interaction with the substrate through a bimolecular quenching process if suitable medium conditions are provided. The EnT process can be described as a simultaneous two-electron exchange mechanism: the donor 4Cz-IPN transfers an electron to the lowest unoccupied molecular orbital (LUMO) of acceptor **1a**, concomitantly receiving an electron from the highest occupied molecular orbital (HOMO) of the acceptor. In a homogeneous sole THF solution, only a 1:1 molar ratio of **1a** (46% yield) to **1a'** (46% yield) was identified, and no desired [2 + 2] photocycloaddition product was discerned. This can be attributed to the rapid dynamics of *E*/*Z* isomerization[27], favoring equilibrium attainment over the progression of dearomative coupling in general organic solutions. In sharp contrast, under on-water conditions, the reactive hypothetical (*E*)-like triplet species actively participate in the formation of exciplexes with aromatic 2,3-benzofuran (**2a**). This interaction is probably induced by the reduced intermolecular distance between the (*E*)-like triplet species and **2a**, which can be attributed to the significant influence of the water-oil interface, resembling high-pressure conditions[2,48]. Plausible evidence for exciplex formation[49] was supported by the red-shifted maximum emission wavelength of normalized photoluminescence (PL), indicating relatively lower energy requirements for the reaction compared to an organic solvent medium (for details, see Supplementary Information). Notably, crude [19]F NMR analysis during the reaction revealed quantitative conversion to the *Z*-isomer **1a'** on a brief timescale (<10 min). Following intermolecular coupling leading to a new C–C bond formation, a 1,4-diradical species appeared, culminating in π–π interaction favored for **3aa**-*exo* (major) and sterically less hindered **3aa**-*endo* (minor) products (Fig. 5E).

## Discussion

In summary, we have developed an on-water accelerated dearomative aquaphotocatalysis for accessing heterocyclic alkyl SuFEx hubs. Water plays a pivotal role in expediting the [2 + 2] cycloaddition between β-arylated ethenesulfonyl fluorides and biorelevant heteroaromatics such as 2,3-benzofuran, 1-benzothiophene, *N*-Boc-protected indole, 3-methylbenzofuran, 3-methylbenzothiophene, and furan, which is facilitated by the high-pressure reactivity amplification effects at the water-oil interface. In conventional organic solvents, pronounced starting material isomerization is predominant, whereas on-water acceleration facilitates the formation of the desired photocycloaddition products. Notably, a gram-scale demonstration and SuFEx ligation with bioactive molecules, such as estrone and stanolone, were showcased. By leveraging its water compatibility, we envision that the developed methodology could be extended to enable the facile modular synthesis of small molecule coupling partners for applications in DNA-encoded libraries (DELs) and antibody-drug conjugates (ADCs) under aqueous conditions.

## Methods

General procedure for dearomative [2 + 2] cycloaddition: β-arylated ESF (0.4 mmol, 1.0 eq.) and 4Cz-IPN (2.0 mol%) were added in a vial without further drying with multiple magnetic stirring bars. Afterwards, the corresponding heteroaromatic compound (2.0 mmol, 5.0 eq.) and $H_2O$ (deionized, 10 L/mol, 4.0 mL) were sequentially added to the reaction vial without further degassing. The suspension was then stirred vigorously (rpm > 1000) under irradiation with a 50 W 456 nm blue LED for 24 h. The distance between the light source and the reaction vial was approximately 5 cm, and an electric fan (PR160 Rig w/ Fan Kit) was used to decrease the temperature of the reaction vial. Subsequently, the crude mixture was extracted with EtOAc (three times)/brine. The organic layer was then dried over anhydrous $Na_2SO_4$ and concentrated under reduced pressure. The resulting residue was purified by column chromatography to afford the desired [2 + 2] cycloadducts, and further purification was performed by recrystallization.

## Data availability

The data supporting the findings of this study are available within the article and its Supplementary Information. Additional data are available from the corresponding author upon request. Crystallographic data for the structures reported in this Article have been deposited at the Cambridge Crystallographic Data Centre, under deposition numbers CCDC 2279317 (**3aa**-*exo*), 2279318 (**3aa**-*endo*), 2279322 (**3da**-*exo*), 2279321 (**3ma**-*exo*), 2295179 (**3ab**-*exo*), 2295180 (**3ac**-*exo*), 2295183 (**3ae**-*endo*).

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

## Acknowledgements

The generous support of the Ministry of Science, ICT and Future Planning of Korea (2020R1C1C1006440, 2019R1A6A1A10073079, RS-2023-00259659, and RS-2023-00219859) and Ministry of Education (National Research Facilities and Equipment Center: 2022R1A6C101A751 and 2022R1A6C102A913) are gratefully acknowledged. This research was supported by the Sungkyunkwan University and the BK21 FOUR (Graduate School Innovation) funded by the Ministry of Education (MOE, Korea) and National Research Foundation of Korea (NRF).

## Author contributions

S.B.K. and H.Y.B. developed the reaction, investigated the substrate scope, derivatizations of the products, and implemented analytical

studies. D.H.K. supported experiments and analyses. H.Y.B. and S.B.K. wrote the manuscript. H.Y.B. designed and oversaw the project.

## Competing interests

The authors declare no competing interests.
