## [Peer Review File · Nature Communications]

REVIEWER COMMENTS

Reviewer #1 (Remarks to the Author):

The authors have developed an “on-water” accelerated dearomative aquaphotocatalysis for accessing a series of heterocyclic alkyl SuFEx hubs. In this study, water is claimed to accelerate the [2+2] cycloaddition between β -arylated ethenesulfonyl fluorides and biorelevant heteroaromatics. Under these conditions, a series of heteroaromatics can react efficiently with ethenesulfonyl fluorides using water as solvent. In addition, the employment of photocycloaddition products for SuFEx ligation with bioactive molecules were showcased. Mechanistic experiments illustrated the “on-water” accelerated aquaphotocatalysis phenomenon. The manuscript is recommended to accept before the following issues to be addressed:

1. The title of the manuscript is confusing. Water is claimed to play a crucial role in this study. Did the authors measure the exact solubility of benzofuran and ethenesulfonyl fluoride in water?
2. Could the red-shift of PL be caused by 4CzIPN in different solutions other than the exciplex formation?
3. Most of the substrates focuses on ethenesulfonyl fluoride and the authors claim that β -unsubstituted ESF were inactive in this catalytic reaction. What about disubstituted or trisubstituted ethenesulfonyl fluorides? In addition, in terms of heteroaromatics, benzofuran derivatives with electron-withdrawing or electron-donating groups, tryptophan derivatives or other electron-rich heterocycles could be tolerated under optimal reaction condition?
4. In the cases of methyl substituted hetero aromatics such as 2d and 2e, endo products were obtained in major form, here needs explanations.
5. It is noticed that the yield of product 9 was only 14%, Have the authors tested other methods such as Moses' conditions (Angew. Chem. Int. Ed., 2022, 61, e202112375)?

Reviewer #2 (Remarks to the Author):

The authors describe 4CzIPN-catalyzed [2+2] photocycloaddition between β -arylated ethenesulfonyl fluoride and benzofurans, benzothiophenes, and N-Boc indole that is promoted by the use of water as a reaction medium. The experiments were performed in an appropriate way and the results are clearly presented. This is interesting observation and I would recommend it for acceptance after minor points shown below are addressed.

In page 1, line 33 the authors describe "remarkable acceleration driven by bulk aqueous media has been absent". We can find some examples that may involve on-water mechanism as follows:

Green Chem. 2019, 21, 2119 (99% vs. up to 63% in organic solvents); Org. Lett. 2015, 17, 884 (83% vs. up to 43% in organic solvents); Chem. Eur. J. 2014, 20, 2960 (35% vs. <20% in organic solvents);

Reviewer #3 (Remarks to the Author):

In this paper, Bae and coworkers describe a [2+2] photoaddition between α,β -unsaturated arylsulfonyl fluoride derivatives and heteroaromatics such as benzofurans, benzothiophenes, indoles, etc. A range of adducts are prepared and well characterized; furthermore, the authors demonstrate the further transformation of the products in SuFEx click-reactions.

The unique part of the paper that merits publication in a high-impact journal like Nature Communications is the fact that the reaction only takes place under light-catalysis "on water" – a novel process the authors named "aquaphotocatalysis". The authors also undertake a series of mechanistic investigations to prove the unique reactivity that aquaphotocatalysis enforces.

Overall, I find the paper well-written and highly interesting! I think the concept of aquaphotocatalysis is of broader interest than the second selling point, i.e. that the products are SuFEx click-reagents of interest to medicinal chemistry and bioactivity. The products do not necessarily have such properties just because they are derived from benzofuran/indole scaffolds. An example of this imbalance between the novel concept of aquaphotocatalysis and the hot, but not so novel, concept of "click-chemistry" is seen in Figure 2B: Here the authors demonstrate that other sulfone derivatives also react as electrophiles (except the sulfonylchloride which is expected due to poor water stability); however, this fact is almost hidden with reference to that the product is not a "click reagent". I would instead lift this point and ideally further explore the substrate scope, e.g. would sulfones, sulfonamides, or even non sulfur-derivatives react as well? Perhaps the topic for follow-up studies.

Given that the work describes on-water transformations of *p*-arylenesulfonylfluorides, I lack one key reference for the substrate preparation. Arvidsson et al described the on-water preparation of *p*-arylenesulfonylfluorides in a paper that was submitted and published some months before the Sharpless paper that was used for preparing the substrates in this work, i.e. *J. Org. Chem.* 2016, 81, 6, 2618–2623 vs. *Angew Chem Int Ed Engl* 2016, 14155-14158. Even if the Sharpless method was used for the substrates here, I consider it respectful to cite also the JOC paper, especially since it also exemplifies on-water conditions in sulfonylfluoride chemistry.

Some minor spelling errors could be corrected before publication, e.g. “aqua” in Fig2B.

In the following, we provide a point-by-point response to the referee comments to the revision of the *Nature Communications* (NCOMMS-23-52650).

REVIEWER COMMENTS

Reviewer #1 (Remarks to the Author):

The authors have developed an “on-water” accelerated dearomative aquaphotocatalysis for accessing a series of heterocyclic alkyl SuFEx hubs. In this study, water is claimed to accelerate the [2+2] cycloaddition between β -arylated ethenesulfonyl fluorides and biorelevant heteroaromatics. Under these conditions, a series of heteroaromatics can react efficiently with ethenesulfonyl fluorides using water as solvent. In addition, the employment of photocycloaddition products for SuFEx ligation with bioactive molecules were showcased. Mechanistic experiments illustrated the “on-water” accelerated aquaphotocatalysis phenomenon. The manuscript is recommended to accept before the following issues to be addressed:

→ We are grateful for the recognition and positive assessment.

We appreciate the time and effort the reviewer has dedicated to providing insightful feedback on ways to strengthen our paper. As the reviewer mentioned, substantial changes have been made based on the constructive advice obtained from the previous submission.

1. The title of the manuscript is confusing. Water is claimed to play a crucial role in this study.

→ We are grateful for the valuable advice.

For clarity, we changed the title as:

“On-Water” Accelerated Dearomative Cycloaddition via Aquaphotocatalysis

Did the authors measure the exact solubility of benzofuran and ethenesulfonyl fluoride in water?

→ We are grateful for the recognition and helpful comments.

To reflect the reviewer’s comment, we investigated extensive solubility experiments of organic components.

1) Initially, for the solubility test, organic components were introduced into water; however, they did not dissolve at all. Pictures are as follows.

(*E*)-phenyl ESF on water

benzofuran on water

benzothiophene on water

Boc-indole on water

2) Extensive NMR experiments were conducted. For the quantitative measurement of solubility, a biphasic solution of $D_2O/CDCl_3$ was prepared, to which (i) (*E*)- β -phenyl ethene-sulfonyl fluoride, (ii) benzofuran, (iii) benzothiophene, or (iv) Boc-indole were added at a concentration of 0.1 mmol each. 1,4-Dimethoxybenzene was added as an internal standard at 0.1 mmol.

(i) (*E*)-phenyl ESF:

¹H NMR measurement sample was conducted using 0.1 mmol of (*E*)-phenyl ESF (**1a**) and 0.1 mmol of 1,4-dimethoxybenzene in biphasic solution (1.0 mL of CDCl₃ and 1.0 mL of D₂O).

This data is a spectrum of CDCl₃ phase.

This data is a spectrum of D₂O phase.

(ii) benzofuran:

^1H NMR measurement sample was conducted using 0.1 mmol of benzofuran (**2a**) and 0.1 mol of 1,4-dimethoxybenzene in biphasic solution (1.0 mL of CDCl_3 and 1.0 mL of D_2O).

This data is a spectrum of CDCl_3 phase.

This data is a spectrum of D_2O phase.

(iii) benzothiophene:

^1H NMR measurement sample was conducted using 0.1 mmol of benzothiophene (**2b**) and 0.1 mmol of 1,4-dimethoxybenzene in biphasic solution (1.0 mL of CDCl_3 and 1.0 mL of D_2O).

This data is a spectrum of CDCl_3 phase.

This data is a spectrum of D_2O phase.

(iv) Boc-indole:

^1H NMR measurement sample was conducted using 0.1 mmol of Boc-indole (**2c**) and 0.1 mmol of 1,4-dimethoxybenzene in biphasic solution (1.0 mL of CDCl_3 and 1.0 mL of D_2O).

This data is a spectrum of CDCl_3 phase.

This data is a spectrum of D_2O phase.

To summarize, organic compounds are dissolved only in CDCl_3 , totally insoluble in D_2O . No peaks of the organic reagents were detected in the D_2O layer.

2. Could the red-shift of PL be caused by 4CzIPN in different solutions other than the exciplex formation?

→ We are grateful for the valuable advice. To address this issue, we conducted extensive experiments.

1) Firstly, a mixed solution of DMSO/water was used: [4Cz-IPN + **1a** (phenyl ESF)] and **2a** (benzofuran) in the various ratio of the media (DMSO + H_2O).

The solutions were prepared 10 mM of **1a** and 40 μM of 4Cz-IPN using:

- (a) 10 mL of anhydrous DMSO only
- (b) 8 mL dried DMSO + 2 mL distilled H_2O
- (c) 6.5 mL dried DMSO + 3.5 mL distilled H_2O
- (d) 5 mL dried DMSO + 5 mL distilled H_2O

Under the mentioned conditions, solutions were prepared and PL was measured: (a) [DMSO only], (b) [DMSO: water = 4:1], (c) [DMSO: water = 2:1], and (d) [DMSO: water = 1:1].

In the case of (d), the solution was heterogeneous (turbid suspension), which posed difficulties for accurate photoluminescence (PL) measurements. Nonetheless, a trend was observed that similar to the behavior in THF/ H_2O media, an increase in the water ratio in DMSO/ H_2O media also tends to result in a red shift. Below are the summarized results of the PL measurements.

■ DMSO ■ water

Fluorescence quenching of excited photocatalyst 4Cz-IPN (40 μM) using **1a** (10 μM), with different concentrations of **2a** (0 mM, 50 mM, 100 mM, 150mM, and 200 mM) under different

media (DMSO/water ratio = (i) 100:0, (ii) 80:20, (iii) 65:35, (iv) normalized PL spectrum of (i), (ii) and (iii).

2) Mixtures of MeOH/water were prepared; however, the photocatalyst 4Cz-IPN precipitated in methanol, rendering accurate measurement of photoluminescence (PL) impossible.

[4Cz-IPN + **1a** (phenyl ESF)] and **2a** (benzofuran) in the various ratio of the media (MeOH + H₂O). The solutions were prepared 10 mM of **1a** and 40 μM of 4Cz-IPN using:

(e) 10 mL of anhydrous MeOH only

(f) 8 mL dried MeOH + 2 mL distilled H₂O

(g) 6.5 mL dried MeOH + 3.5 mL distilled H₂O

(h) 5 mL dried MeOH + 5 mL distilled H₂O

3. Most of the substrates focuses on ethenesulfonyl fluoride and the authors claim that β-unsubstituted ESF were inactive in this catalytic reaction. What about disubstituted or trisubstituted ethenesulfonyl fluorides? In addition, in terms of heteroaromatics, benzofuran derivatives with electron-withdrawing or electron-donating groups, tryptophan derivatives or other electron-rich heterocycles could be tolerated under optimal reaction condition?

→ We are grateful for the valuable advice.

1) To reflect the reviewer's advice, reactions involving benzofuran and benzothiophene heterocycles substituted with either electron-withdrawing groups (EWGs) or electron-donating groups (EDGs) were carried out.

Notably, we successfully expanded the scope to include protected tryptamine, a derivative of tryptophan. In this revision, six new compounds were added, and their structures are presented as follows (see SI for detailed analytical data).

To reflect this issue, we changed substrate scope part as follows (see Fig. 3)

2,3-benzofuran

1-benzothiophene

N-Boc indole

3-methylbenzofuran

3-methylbenzothiophene

furan

tryptophan derivatives

benzofuran derivatives

benzothiophene derivatives

2) The synthesis of 2,2-disubstituted and trisubstituted ethenesulfonyl fluorides requires the use of SO_2F_2 (g), but its utilization was impossible due to the safety and environmental regulations in our laboratory. However, as an alternative, we successfully synthesized **1r**, a bromo-substituted ESF at the alpha position, and its derivative, (*Z*)-1-bromo-2-arylethene-1-sulfonyl fluorides **1s** (Refer to: *Org. Lett.* **2022**, *24*, 4046–4051).

3) Experiments were conducted on the newly synthesized **1r** and **1s** under our aqua-photocatalysis conditions. Unfortunately, the reactions did not proceed in any of the cases. The absence of any observed *E/Z* isomers suggests that the triplet excited energy of the involved substrates likely does not match appropriately with the used catalyst, 4Cz-IPN. We plan to explore further research utilizing ESFs with various substituents as a follow-up study.

4. In the cases of methyl substituted hetero aromatics such as **2d** and **2e**, endo products were obtained in major form, here needs explanations.

→ We are grateful for the valuable advice.

We explain the predominance of the *endo*-product in the [2 + 2] cycloaddition of substrates **2d** and **2e**, which have a substituted 3-position on benzofuran or benzothiophene, based on the related references [*J. Am. Chem. Soc.* **2004**, *126*, 2838–2846; *Acc. Chem. Res.* **2004**, *37*, 919–928]. According to the authors, in the case of 3-methyl furan, “the conformer **T-C2bb'** is a more favorable structure than **T-C1bb'** ($\Delta E_{\text{rel}} = 1.1$ kcal/mol), since the steric repulsion between the methyl and diphenylmethyl group is severer than that between the hydrogen and the diphenylmethyl group. (*J. Am. Chem. Soc.* **2004**, *126*, 2838–2846.)”

This explanation is based on the energies and steric interactions within the molecular structures involved.

Similarly, in our reaction as well, with the 3-methylated heterocyclic substrates **2d** and **2e** (right scheme), the presence of steric repulsion between the methyl group and the phenyl group of **1a** would favor the formation of an intermediate in the form of **T-C2bb'**. Conversely, for **2a**, which lacks substitution at the 3-position, is anticipated to form an intermediate resembling the structure of **T-C1bb'** (left scheme).

We revised the main text as follows and added relevant reference.

“Different heteroaromatic coupling partners **2**, such as 1-benzothiophene (**2b**), *N*-Boc-protected indole (**2c**), 3-methylbenzofuran (**2d**), 3-methylbenzothiophene (**2e**), furan (**2f**), *N*-Boc-protected-L-tryptophan derivative (**2g**), functionalized benzofuran derivatives (**2h–2j**), and functionalized benzothiophene derivatives (**2k** and **2l**) were smoothly converted into their respective products (**3ab–3al**) with yields of up to 99%. Interestingly, it is worth to note that in the cases of alkyl substituted heteroaromatics at 3-position such as **2d**, **2e**, and **2g**, *endo* products were obtained in major form (**3ad**, *exo* : *endo* = 15 : 85; **3ae**, *exo* : *endo* = 21 : 79; **3ag**, *exo* : *endo* = 44 : 56). The reason is unclear at this stage, however, the presence of steric repulsion between the alkyl group at the 3-position of heteroaromatics **2** and the phenyl group of **1a** is believed to favor *endo*-product formation (this is in accordance with a similar situation reported in the Paternò-Büchi reaction, where the intermolecular steric repulsion between the alkyl group on the heteroaromatic and the bulky group on the counterpart was identified as the key factor).⁴¹”

5. It is noticed that the yield of product **9** was only 14%, Have the authors tested other methods such as Moses' conditions (Angew. Chem. Int. Ed., 2022, 61, e202112375)?
 → We are grateful for the valuable advice.

In accordance with the suggested reference, deviations from the optimized conditions were tested on the reaction using **3aa-exo** at a 0.2 mmol scale. The results are as follows.

When BTMG (2-tert-butyl-1,1,3,3-tetramethylguanidine) was utilized as the base catalyst, the isolated yield did not surpass approximately 11%, indicating no significant improvement. However, fortunately, by employing DBU (1,8-Diazabicyclo[5.4.0]undec-7-ene, 1.0 eq.), we obtained more than double our initial results, increasing the yield from 14% to 31%.

Entry	deviation	isolated yield
1	DBU 1.0 eq. as base in 60 °C, 24 h	31%
2	BTMG 20 mol% as base in 25 °C, 1 h	trace
3	BTMG 20 mol% as base in 25 °C, 24 h	trace
4	BTMG 20 mol% as base in 60 °C, 1 h	5%
5	BTMG 20 mol% as base in 60 °C, 24 h	11%

To reflect these results and revise the reaction conditions, Fig. 4 was changed accordingly. Related experimental results have been added to the Supplementary Information.

■ **Reaction conditions:**

(i) benzylamine (2.0 equiv.), Et₃N (2.0 equiv.), MeCN (0.4 M), 60 °C, 24 h

(iii) phenylacetylene (1.1 equiv.), CuTC (10 mol%), PhMe (0.05 M), r.t., 24 h

(v) estrone (1.2 equiv.), Cs₂CO₃ (2.0 equiv.), MeCN (0.2 M), r.t., 24 h
then Cs₂CO₃ (0.5 equiv.) r.t., 12 h

(ii) TMSN₃ (1.5 equiv.), DMAP (1.5 equiv.), MeCN (0.5 M), 60 °C, overnight

(iv) 4-tert-butylphenol (1.1 equiv.), Cs₂CO₃ (2.0 equiv.), MeCN (0.2 M), r.t., 12 h

(vi) stanolone (1.2 equiv.), HMDS (1.0 equiv.), DBU (1.0 equiv.), MeCN (0.4 M), 60 °C, 24 h

Reviewer #2 (Remarks to the Author):

The authors describe 4CzIPN-catalyzed [2+2] photocycloaddition between β -arylated ethenesulfonyl fluoride and benzofurans, benzothiophenes, and N-Boc indole that is promoted by the use of water as a reaction medium. The experiments were performed in an appropriate way and the results are clearly presented. This is interesting observation and I would recommend it for acceptance after minor points shown below are addressed.

→ We are grateful for the recognition and positive assessment.

We appreciate the time and effort the reviewer has dedicated to providing insightful feedback on ways to strengthen our paper. As the reviewer mentioned, substantial changes have been made based on the constructive advice obtained from the previous submission.

In page 1, line 33 the authors describe "remarkable acceleration driven by bulk aqueous media has been absent". We can find some examples that may involve on-water mechanism as follows:

Green Chem. 2019, 21, 2119 (99% vs. up to 63% in organic solvents); Org. Lett. 2015, 17, 884 (83% vs. up to 43% in organic solvents); Chem. Eur. J. 2014, 20, 2960 (35% vs. <20% in organic solvents);

→ We are grateful for the valuable advice.

The references provided demonstrate photocatalytic reactions that differ somewhat from our identified results.

1) *Green Chem.* **2019**, *21*, 211.:

"These results clearly showed the superiority of the proton reduction catalyst **IV**, because the Co-oxime complex (**IV**) is more soluble in water."

- Comment: dehydrogenation of cyclic amines using a water-soluble metal catalyst.

2) *Org. Lett.* **2015**, *17*, 884 (83% vs. up to 43% in organic solvents);

- Comment: similar water-soluble metal catalysis for cross-dehydrogenative coupling.

3) *Chem. Eur. J.* **2014**, *20*, 2960 (35% vs. <20% in organic solvents); "Water appears to be an ideal solvent for reactions of this nature, as both the substrates and catalyst are soluble and stable in aqueous solution. The solubility of a substrate in water is crucial to the reaction yield."

- Comment: it is clearly stated that all substrates are soluble in aqueous solution.

There are no known instances of accelerated reactions involving 'water-insoluble' reaction partners and hydrophobic organic photocatalysts. To clarify, we revised the relevant sentence as follows:

"However, remarkable acceleration of cycloadditions driven by bulk aqueous media employing a combination of water-insoluble substrates and hydrophobic photocatalysts has been absent in visible-light photocatalysis."

Reviewer #3 (Remarks to the Author):

In this paper, Bae and coworkers describe a [2+2] photoaddition between a,b-unsaturated arylsulfonylfluoride derivatives and heteroaromatics such as benzofurans, benzothiophenes, indoles, etc. A range of adducts are prepared and well characterized; furthermore, the authors demonstrate the further transformation of the products in SuFEx click-reactions.

The unique part of the paper that merits publication in a high-impact journal like Nature Communications is the fact that the reaction only takes place under light-catalysis “on water” – a novel process the authors named “aquaphotocatalysis”. The authors also undertake a series of mechanistic investigations to prove the unique reactivity that aquaphotocatalysis enforces.

Overall, I find the paper well-written and highly interesting!

→ We are grateful for the recognition and positive assessment.

We appreciate the time and effort the reviewer has dedicated to providing insightful feedback on ways to strengthen our paper.

I think the concept of aquaphotocatalysis is of broader interest than the second selling point, i.e. that the products are SuFEx click-reagents of interest to medicinal chemistry and bioactivity. The products do not necessarily have such properties just because they are derived from benzofuran/indole scaffolds. An example of this imbalance between the novel concept of aquaphotocatalysis and the hot, but not so novel, concept of “click-chemistry” is seen in Figure 2B: Here the authors demonstrate that other sulfone derivatives also react as electrophiles (except the sulfonylchloride which is expected due to poor water stability); however, this fact is almost hidden with reference to that the product is not a “click reagent”. I would instead lift this point and ideally further explore the substrate scope, e.g. would sulfones, sulfonamides, or even non sulfur-derivatives react as well? Perhaps the topic for follow-up studies.

→ We are grateful for the recognition and positive assessment.

As specified in the manuscript, the reaction proceeds very well when substituting -SO₂Ph for -SO₂F. As the reviewer pointed out, we tested reactions involving non-sulfur derivatives, such as nitroolefins incorporating the -NO₂ group, achieving successful preliminary experimental results. These will be the subjects of our next research focus.

Based on the reviewer's advice, we were curious about the reactivity of functional olefins, which have undergone phenolic donor and SuFEx ligation, in [2+2] photocycloaddition reactions. We conducted extensive experiments as follows.

Initially, we synthesized compound **1t** via the SuFEx reaction and then subjected it to our aqua-photocatalytic conditions. In this case, the reaction did proceed, yielding the product with a 71% yield, resulting in *exo/endo* isomers at a ratio of 81:19. The separation of the product in this process proved highly difficult. Hence, we concluded that proceeding with a catalytic [2+2] photocycloaddition is significantly more advantageous regarding product yield.

When using phenol as the donor, the desired products form an *exo*-/*endo*- mixture, similar to compound **7**, and the positions of the isomers on the TLC are so close that separation by column chromatography was difficult.

Similarly, when using benzylamine, the formation of *exo/endo* isomers, which are very challenging to separate, was observed (75:25, 61% yield). In this instance as well, we concluded that proceeding with a catalytic [2+2] photocycloaddition is significantly more advantageous in terms of product yield.

Given that the work describes on-water transformations of *b*-arylethenesulfonylfluorides, I lack one key reference for the substrate preparation. Arvidsson et al described the on-water preparation of *b*-arylethenesulfonylfluorides in a paper that was submitted and published some months before the Sharpless paper that was used for preparing the substrates in this work, i.e. *J. Org. Chem.* 2016, 81, 6, 2618–2623 vs. *Angew Chem Int Ed Engl* 2016, 14155–14158. Even if the Sharpless method was used for the substrates here, I consider it respectful to cite also the JOC paper, especially since it also exemplifies on-water conditions in sulfonylfluoride chemistry.

→ We are very grateful for the considerate advice. We agreed with this comment and added the relevant reference (*J. Org. Chem.* 2016, 81, 6, 2618–2623) in the main text.

Some minor spelling errors could be corrected before publication, e.g. “aqua” in Fig2B.

→ We are grateful for the constructive advice. We could not find the term “aqua” in Fig 2B. However, we have made corrections to typographical errors throughout the manuscript.

REVIEWERS' COMMENTS

Reviewer #1 (Remarks to the Author):

All the concerns have been addressed and the revised manuscript is recommended to publish.

Reviewer #3 (Remarks to the Author):

The authors have made a large effort to satisfy all reviewers by undertaking several new experiments. The outcome of this effort was an even better manuscript and many ideas for follow-up work.

I recommend to publish the paper as is.